# TCEPVDB: Artificial Intelligence-Based Proteome-Wide Screening of Antigens and Linear T-Cell Epitopes in the Poxviruses and the Development of a Repository

**DOI:** 10.3390/proteomes13040058

**Published:** 2025-11-06

**Authors:** Mansi Dutt, Anuj Kumar, Ali Toloue Ostadgavahi, David J. Kelvin, Gustavo Sganzerla Martinez

**Affiliations:** 1Department of Microbiology and Immunology, Canadian Center for Vaccinology (CCfV), Faculty of Medicine, Dalhousie University, Halifax, NS B3K 6R8, Canada; mansidutt@dal.ca (M.D.); kumaranuj@dal.ca (A.K.); ali.toloue@dal.ca (A.T.O.); gustavo.sganzerla@dal.ca (G.S.M.); 2Laboratory of Immunity, Shantou University Medical College, Jinping, Shantou 515041, China; 3BioForge Canada Limited, Halifax, NS B3N 3B9, Canada

**Keywords:** reverse vaccinology, poxviruses, repository, vaccine development

## Abstract

Background: Poxviruses constitute a family of large dsDNA viruses that can infect a plethora of species including humans. Historically, poxviruses have caused a health burden in multiple outbreaks. The large genome of poxviruses favors reverse vaccinology approaches that can determine potential antigens and epitopes. Here, we propose the modeling of a user-friendly database containing the predicted antigens and epitopes of a large cohort of poxvirus proteomes using the existing PoxiPred method for reverse vaccinology of poxviruses. Methods: In the present study, we obtained the whole proteomes of as many as 37 distinct poxviruses. We utilized each proteome to predict both antigenic proteins and T-cell epitopes of poxviruses with the aid of an Artificial Intelligence method, namely the PoxiPred method. Results: In total, we predicted 3966 proteins as potential antigen targets. Of note, we considered that this protein may exist in a set of proteoforms. Subsets of these proteins constituted a comprehensive repository of 54,291 linear T-cell epitopes. We combined the outcome of the predictions in the format of a web tool that delivers a database of antigens and epitopes of poxviruses. We also developed a comprehensive repository dedicated to providing access to end-users to obtain AI-based screened antigens and T-cell epitopes of poxviruses in a user-friendly manner. These antigens and epitopes can be utilized to design experiments for the development of effective vaccines against a plethora of poxviruses. Conclusions: The TCEPVDB repository, already deployed to the web under an open-source coding philosophy, is free to use, does not require any login, does not store any information from its users.

## 1. Introduction

Poxviruses represent a family of large dsDNA viruses [1] whose genome encodes several proteins that are key for driving the virus pathogenicity and transmissibility. Poxviruses can infect invertebrates and vertebrates, including homo sapiens. Throughout history, poxviruses, in the form of the variola virus (VARV), which causes the disease smallpox, have been listed among the greatest infectious killers of mankind [2]. Archaeological cues of VARV can be traced back to ancient Egypt in 1157 BC, when evidence found in the mummified body of the pharaoh Ramses V suggested smallpox-derived lesions. Although the smallpox was declared as an eradicated disease in 1980, other poxviruses still pose a threat to human life [3]. In May 2022, when the world was still in the SARS-CoV-2 pandemic, sustained human-to-human transmission of the mpox virus (MPXV) started being reported in areas in which the disease, mpox, is not endemic, having resulted in 93,497 cases (91,373 in locations that have not historically reported mpox) and 177 deaths (156 in locations that have not historically reported mpox) worldwide [4]. Apart from MPXV, other poxviruses such as tanapox [5], orf [6], and molluscum contagiosum [7] can also infect humans. Poxviruses also pose a veterinary threat to several animals. Lumpy Skin Disease Virus (LSDV) affects cattle with mouth ulcers that can cause weakness, loss of appetite, and reduced milk production. Camelpox virus accounts for increased weight loss, reduced milk production, and mortality in camelids [8]. Birds are also affected by poxviruses, as approximately 9000 bird species 232 have been reported to have acquired a natural poxvirus infection [9].

One of the components of an immune response is cellular immunity or T-cell-mediated immunity, responsible for defending one’s body against intracellular pathogens. At the start, the cells of pathogens can express antigens, which are proteins that are recognized by T-cell receptors (TCRs). These antigens might contain specific regions called epitopes, which are short protein subsequences that directly interact with TCRs. The epitopes are presented by major histocompatibility complex (MHC) molecules on the surface of cells that present antigens. After recognizing the epitope–MHC complex, the specific T cells are activated and cloned, leading to the proliferation of antigen-specific T cells adapted to present an immune response against a specific antigen of a pathogen. Upon activation, T cells can specialize in different functions such as antibody production, inducing apoptosis in target cells, suppressing excessive immune response, and generating long-lasting immunological memory [10].

With whole-genome sequencing techniques having become accessible, the genome of pathogens can be explored to determine the potential antigenic repertoire of an organism from its genomic sequence; this process is termed ‘reverse vaccinology’ [11,12]. The dissemination of Artificial Intelligence (AI) has enabled the analysis and prediction of high volumes of genomic data [13,14]. Examples of AI techniques have also been employed as a key step in the analysis and classification of data in the reverse vaccinology paradigm [15,16]. Examples of vaccines that are in the market as of August 2025 include a vaccine for the influenza H5N1 virus [17]; Bexsero, a multicomponent meningococcal serogroup B (4CMenB) vaccine [18]; and Shingrix, a vaccine developed against Shingles, which is caused by the Varicella Zoster virus [19].

In this work, we present the T-Cell Epitopes Poxviruses Database (TCEPVDB). We obtained the protein repertoire of 37 distinct poxviruses and submitted them for vaccine components prediction using the PoxiPred [20] method. The predicted outputs are designed in a user-friendly database. Here, we document the development stage of TCEPVDB. Users interested in exploring the data of TCEPVDB can freely access the tool at https://tcepvdb.microbiologyandimmunology.dal.ca (accessed on 11 October 2025).

## 2. Materials and Methods

A flowchart of the pipeline utilized in the present study is illustrated in Figure 1.

### 2.1. Obtention of Protein Répertoire

We obtained the protein repertoire of 37 poxviruses. Our search included the query term on NCBI ‘poxvirus’. A total of 49 results matched our query, from which we could obtain a complete genome for 37 distinct poxviruses. When available, we opted to use RefSeq genomes.

### 2.2. Predicting Antigens and LTCEs

PoxiPred [20] was originally developed as an agnostic classification framework for predicting antigens and LTCEs in poxvirus protein datasets, functioning as an early data curation step. For the construction of TCEPVDB, we did not develop new models; instead, we employed the pre-trained Deep Learning Artificial Neural Network (DL-ANN) models for (i) antigen prediction and (ii) LTCE prediction. The models are publicly available at https://github.com/gustavsganzerla/poxipred (accessed on 21 September 2025). Using these existing models, we analyzed the protein repertoire of 37 distinct poxviruses. First, the antigen prediction model was applied to each individual protein from the dataset. Proteins predicted as potential antigens were then fragmented into smaller peptides, which were subsequently evaluated using the LTCE prediction model. In both instances, predictions were considered as positive when the sigmoid output layer of the corresponding pre-trained model produced a score ≥ 0.5.

### 2.3. Web Development

We implemented TCEPVDB as a webtool using the Django framework (version 4.2.4) for Python (version 3.11) web development. First, a host server was purchased in the region of Toronto (Canada). This server was subsequently configured to execute tasks related to the synchronization of a Git (version 2.39.2) repository. The repository serves as the source of version control, with commits being actively contributed over time. We used the relational structured database system SQLite, built in the Django framework. In addition, the deployment architecture incorporated the use of the Apache2 HTTP server to manage proxy functionalities to a domain obtained within Dalhousie University. Apache serves as a reverse proxy to handle requests and direct to the appropriate backend services.

### 2.4. Unified Modeling Language (UML) Artifacts

We used UML to document the development stages of TCEPVDB. First, an entity relationship (ER) diagram to document the logical structure of our database system. Next, an activity diagram was used to visualize the dynamic aspects of the system, representing a user using TCEPVDB to query antigens and epitopes. Both diagrams were developed using the webtool LucidChart for diagramming (Lucid Software, South Jordan, UT, USA).

### 2.5. Conservation of Epitopes Across Poxvirus Species

Predicted epitope sequences from each organism were extracted from their files and scanned against the proteome of all other organisms. For each source organism, the number of its epitopes present in each proteome was counted and compiled into a matrix. The resulting matrix was visualized as a heatmap to illustrate epitope conservation and cross-species distribution.

## 3. Results

### 3.1. Organisms and Predictions

We assessed a total of 7185 proteins from 37 distinct poxviruses. Firstly, each protein was submitted to PoxiPred’s antigen predictor. A total of 3966 proteins were flagged as potential antigens (Table 1). Importantly, these proteins may exist in different proteoforms and may arise from sequence variations, post-translational modifications, etc., which may significantly influence epitope accessibility and immune recognition.

Next, we queried each protein and extracted subsequences of it to submit to the T-cell epitope prediction model. First, we explored the training data of PoxiPred and obtained the interquartile length of the epitopes used for training the model, i.e., the range of 9 to 13 amino acids (Appendix A). To mathematically represent our epitope search, let α be an antigen sequence. Let n be a random variable representing the length of the epitope such that *n* ∈ {9, 10, 11, 12, 13}. The epitope *a*_1_, a_1_ + n is a substring of α starting at position *a*_1_ and ending at position *a*_1_ + *n* − 1, where*epitope*_α1,α1+n_ = a_α1:α1+n−1_(1)

Upon extracting the epitope *a*_1_, *a*_1*+n*_*,* let *a* be updated asα = α_α1+n_(2)
and repeated until length (α) ≤ 13.

In total, we predicted 54,291 LTCEs.

### 3.2. Webtool Functionalities

Users who access TCEPVDB can search in two distinct tables: (i) antigens, which return to the user proteins predicted as antigens, and (ii) epitopes, which return to the user peptides predicted as epitopes. The search term, the only input required from the user, can be of two natures: (i) a search organism, in which epitopes or antigens are searched based on the name or partial name of an organism, and (ii) a search antigen, in which epitopes or antigens are searched based on the name or partial name of a protein. TCEPVDB is implemented using two relational tables, i.e., protein and epitope tables, in an SQLite database. One protein can have 0 to many epitopes associated with it. We modeled the protein and epitope tables as part of an entity relationship diagram (Figure 2A).

Moreover, we modeled a user performing a query operation to TCEPVDB as part of an activity diagram (Figure 2B). First, a user needs to submit a query term, whether in the name of an organism or the name of a protein. Next, the user selects the table to perform the query, i.e., antigens table or epitopes table. The system will then search the query term in the specified table. Finally, the system will either display the results of the submitted query or display a message indicating no results were found.

Upon rendering the results page, the user can click on a ‘download’ icon. If queried to the antigen table, the user will download a .fasta file containing all the proteins predicted as antigens. If queried to the epitopes table, a .fasta file containing the protein of origin followed by all epitopes predicted separated by a tab delimiter will be prepared for the user to download.

TCEPVDB also includes a functionality to display all organisms available in the DB. In the navigation bar, users can select the item ‘All Organisms’. A table similar to Table 1 will be displayed, in which each row represents an organism. The information of an organism is divided into three levels: protein repertoire, antigen, and epitope. In each level, the number of proteins, proteins predicted as antigen, and total number of LTCEs is displayed. On the side of each level, a ‘view’ and a ‘download’ icon is also displayed. Upon clicking in the ‘view’ icon, the user will be directed to another table showing, according to the level of the click, (i) all the proteins of an organism; (ii) all the proteins predicted as antigens of an organism; or (iii) all the epitopes of an organism. Upon clicking the ‘download’ icon, the system will automatically download, according to the level of the click, (i) all proteins of an organism compiled in a .fasta file; (ii) all the proteins predicted as antigens of an organism, compiled in a .fasta file; or (iii) all the LTCEs of an organism, compiled in a .fasta file with the header containing the protein of origin followed by all epitopes predicted separated by a tab delimiter. Both the epitope and antigen models in TCEPVDB contain a score attribute, which is defined by the output of the sigmoid function applied at the final layer of the neural networks implemented in PoxiPred [20]. Entries displayed in TCEPVDB correspond to predicted antigens and epitopes with sigmoid scores ≥ 0.5, reflecting the classification threshold used to determine positive predictions.

Finally, when a user is visualizing either the antigen or the epitope table (resulting from viewing a query or all organisms), we include a url in the ‘Description’ column of the rendered table. The ‘Description’ column contains information on the antigen or the molecular parent of a predicted epitope. Upon clicking the url, the user is directed to the individual view of the antigen, in which a html page is rendered showing the description (Figure 3A); the protein sequence (Figure 3B); a button to download the epitopes predicted for the protein in question (Figure 3C) in .fasta format; and a table containing all the epitopes predicted for that antigen (Figure 3D), containing the column’s epitope number, epitope sequence, epitope prediction score, and genomic coordinates for the start and end of the epitope within the protein. A comparison between newly developed TCEPVDB and other existing vaccine tools and platforms (IEDB [21], Vaxign [22], and VaxiJen [23] is given in Table 2.

### 3.3. Conservation of Predicted Epitopes Across Poxvirus Species

To check the conservation of the predicted epitopes, we investigated the presence or absence of each of the predicted epitopes across the entire protein repertoire of the 37 poxviruses we have data available for at TCEPVDB (Figure 4). The poxviruses ectromelia, horsepox, Mpox, taterapox, vaccinia, variola, camelpox, and cowpox had over one thousand epitopes shared between themselves, attesting for the conservation between these poxviruses which might be exploited in cross-protective vaccine development.

## 4. Discussion

Here, we developed TCEPVDB, a database consisting of AI-predicted antigens and LTCEs from 37 distinct poxviruses that infect humans and animals. TCEPVDB is freely accessible, does not require any login, and does not store any information from its users. Moreover, we followed an open-data policy in developing our tool. It is of our belief that TCEPVDB constitutes an important milestone in the reverse vaccinology paradigm; as of August 2025, no databases specifically focused on providing resources for studying the development of vaccines following reverse vaccinology are available.

The first vaccine whose development was based on the paradigm of reverse vaccinology was proposed by Sette and Rappuoli (2010) [12]. Since then, several applications have become promptly available to facilitate the execution of the reverse vaccinology pipeline [24,25,26]. Despite the absence of a single comprehensive database focused on reverse vaccinology elements, there are resources that, when combined, might assist in predicting antigens and epitopes to facilitate vaccine development. Despite examples such as the Immune Epitope Database (IEDB) [21], Vaxign [22], and Pathosystems Resource Integration Center (PATRIC) [27] not specifically focusing on resources for reverse vaccinology, they all contain valuable data and tools that can be used in the process of identifying potential vaccine targets.

The current iteration of TCEPVDB contains limitations. First, the database is entirely focused on poxviruses. There is compelling evidence to ascertain that true antigens and true epitopes have a conserved profile when compared to proteins/peptides not able to drive an immune response [28,29,30]. Next, we also elicit that TCEPVDB is entirely developed upon T-cell epitopes, which constitute only a share of an immune response. However, the lack of experimentally verified B-cell epitopes hindered the obtention of an inclusive training dataset for constituting a pipeline focused on predicting B-cell epitopes of poxviruses. Finally, the predictions we obtained with PoxiPred [20] and publicized in TCEPVDB are still raw outputs of the PoxiPred pipeline. There are filters that test proteins and epitopes for allergenicity [31] and toxicity [32] of antigens/epitopes that can be employed and will consequently reduce the number of available antigens/epitopes in our database.

The methodology for the prediction of antigens and epitopes utilized in the present study is solely based on the PoxiPred method with ROC AUC metrics, which has been defined as a key indicator of classification performance [20]. Similar trends have already been followed in other related studies. Previously, Souza et al. [33] significantly applied the precision–recall trade-offs to validate model reliability in drug-likeness prediction for the screening of SARS-CoV-2 inhibitors. In another recent study, Kacen et al. [34] highlighted the importance of antigenic landscape modeling with the aid of tumor immunopeptidomics. To predict effective vaccine candidates for mpox or other contagious poxviruses, a combined sequence- and structure-based approach is required to enhance accuracy and productivity, as reported by Pritam [35] in a proteome-wide immunoinformatics study. These reports underscore the value of rigorous evaluation of available epitope prediction tools. In the future, incorporating the different ML-based metrics into the validation of antigen and epitopes embedded in the TCEPVDB may give strength to its comparative positioning and significant utilization for the development of effective vaccine candidates against multiple poxviruses.

In conclusion, the *in silico* step of reverse vaccinology serves primarily as a data curation step, generating preliminary predictions of candidate epitopes and antigens. Their prediction requires subsequent experimental validation and biological interpretation to refine the set of candidates with true potential for incorporation into vaccine designs. 

## 5. Conclusions

The developed TCEPVDB is devoted to providing a comprehensive catalog of a total of 3966 proteins as potential antigen targets and 54,291 linear T-cell epitopes from 37 distinct poxviruses. The antigen proteins and linear T-cell epitopes embedded in this database are predicted using the AI-based PoxiPred method. TCEPVDB is a user-friendly database and can be freely accessed using the following URL: https://tcepvdb.microbiologyandimmunology.dal.ca/ (accessed on 11 October 2025). With further progress in genome sequencing and the AI-based screening of antigens and epitopes, we anticipate that the number of entries in TCEPVDB will eventually grow in the upcoming years. Taken together, the information available in TCEPVDB can be used in efforts of reverse vaccinology, facilitating the rapid development of effective vaccines to tackle poxviruses in a significant manner.

## Figures and Tables

**Figure 1 proteomes-13-00058-f001:**
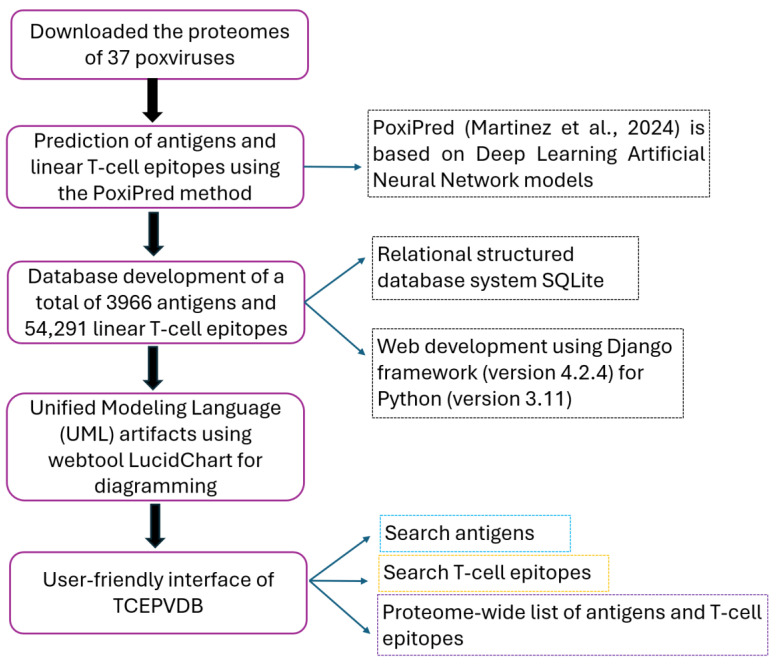
The flowchart of the pipeline utilized in the present study to predict the antigens and linear T-cell epitopes, followed by the development of a comprehensive TCEPVDB. The PoxiPred [20] method was utilized to predict the antigens and T-cell linear epitopes

**Figure 2 proteomes-13-00058-f002:**
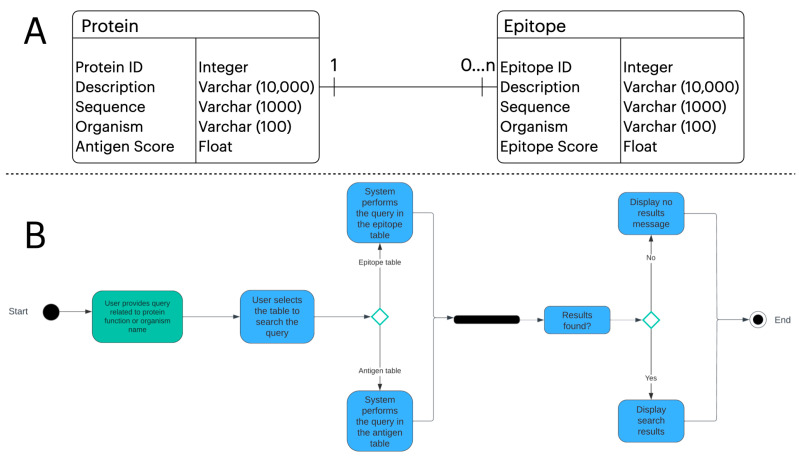
In (**A**), we show the UML entity relationship diagram of the tables modelled in TCEPVDB as SQL objects with their specific attributes and types of variables assigned in the database modelling. There are two tables, antigen and epitope. One antigen can be associated with 0 or many epitopes. In (**B**), we show the UML activity diagram of the workflow of a query to TCEPVDB. A query can be done either in the antigen or epitope model. The modelling of Figure 2 was done in the LucidChart diagramming tool.

**Figure 3 proteomes-13-00058-f003:**
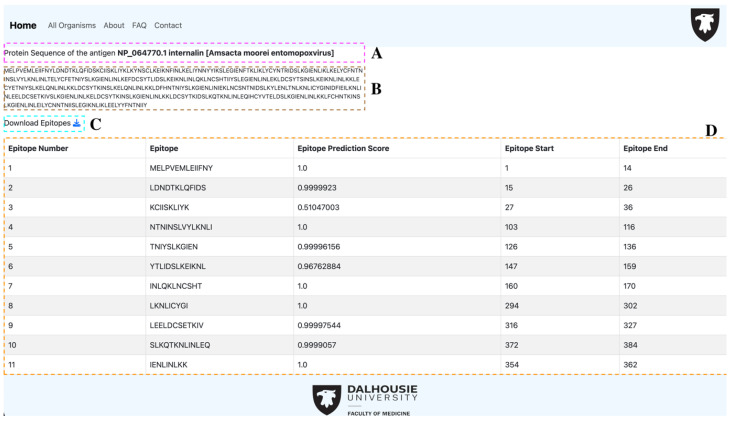
View of an individual antigen and its associated epitopes in TCEPVDB: (**A**) Description of the protein entry; (**B**) Amino acid sequence of the protein; (**C**) Download option to save the results in the .csv format; and (**D**) Tabular representation of the epitopes result.

**Figure 4 proteomes-13-00058-f004:**
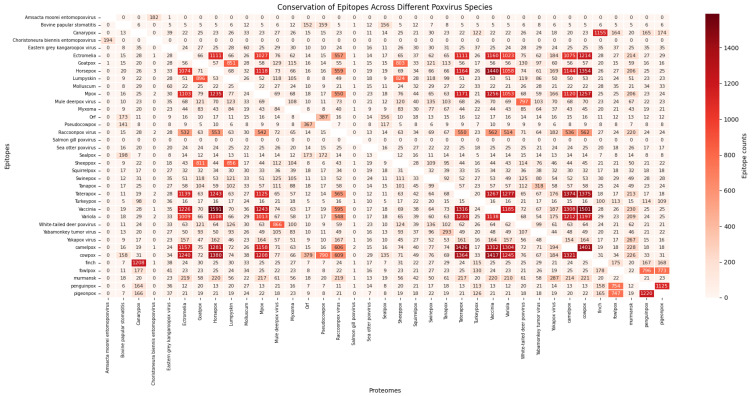
Conservation of epitopes across different poxvirus species. In Figure 4, we show a heatmap of epitope conservation across different poxvirus species. Each epitope that was predicted by PoxiPred and modeled on TCEPVDB had its presence/absence in the proteome of the poxvirus species considered in this study compiled. The heatmap consists of the absolute presence of epitopes across poxvirus species.

**Table 1 proteomes-13-00058-t001:** Protein repertoire, predicted antigens, and predicted LTCEs available in TCEPVDB.

Organism	Genome Accession	Proteins (*n*)	Predicted Antigens (*n*)	Predicted Linear T-Cell Epitopes (*n*)
Amsacta moorei entomopox virus	GCF_000837185.1	294	157	1891
Bovine papular stomatitis virus	GCF_000844045.1	130	61	932
Canarypox virus	GCF_000841685.1	322	167	2387
Choristoneura biennis entomopoxvirus	GCF_000909015.1	334	179	2341
Eastern grey kangaroopox virus	GCF_006450915.1	162	82	1167
Ectromelia virus	GCF_000841905.1	180	127	1714
Goatpox virus	GCF_000840165.1	149	68	1092
Horsepox virus	GCF_000860085.1	228	154	1844
Lumpy skin disease virus	GCF_000839805.1	156	77	1193
Molluscum contagiosum virus	GCF_000843325.1	163	59	895
Mpox virus	GCF_000857045.1	183	134	1777
Mule deerpox virus	GCF_000861985.1	169	81	1113
Myxoma virus	GCF_000843685.1	158	73	924
Orf virus	GCF_000844845.1	130	52	737
Pseudocowpox virus	GCF_000886295.1	125	55	775
Raccoonpox virus	GCF_001029045.1	207	128	1750
Salmon gill poxvirus	GCF_001271235.1	210	105	1490
Sea otter poxvirus	GCF_003260795.1	132	70	1074
Sealpox virus	GCF_002219465.1	119	52	820
Sheeppox virus	GCF_000840205.1	147	72	1113
Squirrelpox virus	GCF_000913615.1	141	63	1083
Swinepox virus	GCF_000839965.1	146	75	1183
Tanapox virus	GCF_000847185.1	155	79	1140
Taterapox virus	GCF_000869985.1	220	140	1761
Turkeypox virus	GCF_001431935.1	170	92	1400
Vaccinia virus	GCF_000860085.1	214	150	1851
Variola virus	GCF_000859885.1	211	142	1663
White-tailed deer poxvirus	MF966153	171	85	1167
Yaba monkey tumor virus	GCF_000845705.1	140	59	902
Yokapox virus	GCF_000892975.1	186	105	1410
Camelpox virus	GCF_000839105.1	261	164	1887
Cowpox virus	GCF_000839185.1	214	142	1959
Finch poxvirus	OM869482	335	186	2414
Fowlpox virus	GCF_000838605.1	251	146	1973
Murmansk poxvirus	GCF_002270885.1	206	115	1647
Penguinpox virus	GCF_000923135.1	242	137	1914
Pigeonpox virus	GCF_000922075.1	224	133	1908

**Table 2 proteomes-13-00058-t002:** A comparison of the TCEPVDB and other existing popular vaccine design tools and platforms.

Name	Focus Organism	Type	Antigen Prediction Method	Epitope Type	Specificity to Poxviruses	Structural Integration	Output Option	User Interface
TCEPVDB	Poxviruses (n = 37)	Database (epitopes + antigen repository)	PoxiPred (ML-based, proteome-wide)	Linear T-cell epitopes	Yes	No, but output can be utilized for structure modeling	Tabular + downloadable .fasta	Web-based, with custom search
IEDB Epitope Tools	Broad (4700+ species)	Prediction + curated experimental database	Multiple ML-based tools (e.g., NetMHCpan)	T-cell, B-cell, MHC ligands	No, but information on multiple poxviruses can be extracted	Partial (some 3D epitope mapping)	Epitope lists, binding scores, and plots	Web-based, modular tools
Vaxign	Broad (bacteria, viruses, parasites)	Pipeline + database	Reverse vaccinology (genomic + ML filters	T-cell (MHC I/II), B-cell	No	Partial (subcellular localization)	Ranked antigen list + epitope predictions based on the scores	Web-based, form-driven
VaxiJen	Broad (pathogen-agnostic)	Standalone prediction tool	Alignment-independent auto cross-covariance	No, only antigenicity scores	Can be used to predict antigenicity scores for individual proteins of poxviruses	No	Antigenicity score	Web-based, simple input

## Data Availability

The original contributions presented in this study are included in the article/Appendix A. Further inquiries can be directed to the corresponding author.

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
