# Peer review of "TCEPVDB: Artificial Intelligence-Based Proteome-Wide Screening of Antigens and Linear T-Cell Epitopes in the Poxviruses and the Development of a Repository"

_proteomes, 2025, doi:10.3390/proteomes13040058_

Round 1

Reviewer 1 Report

Comments and Suggestions for Authors

The manuscript entitled “TCEPVDB: Artificial intelligence based proteome wide screening of antigens and linear T-cell epitopes in the poxviruses and the development of a repository” describes the development of a novel database dedicated to poxvirus antigens and epitopes. The authors employed the PoxiPred deep learning framework to systematically screen the proteomes of 37 distinct poxviruses, predicting 3,966 antigens and more than 54,000 linear T-cell epitopes. These predictions were then organized into a comprehensive, open-access repository TCEPVDB that can be freely queried by the scientific community. The work is timely, highly relevant, and represents a significant step forward in applying computational vaccinology to viral pathogens of public health concern.

The article is well structured and clearly written. The introduction provides a strong rationale for the work, situating poxviruses in both historical and modern contexts, from smallpox eradication to the mpox outbreak of 2022. The authors explain the importance of reverse vaccinology and the role of computational methods in accelerating antigen discovery. This sets the stage well for the contribution of TCEPVDB as a specialized resource built specifically for poxviruses. The methods section is transparent and rigorous. The authors describe in detail how proteomes were retrieved from NCBI, how antigens and epitopes were predicted using the PoxiPred DL-ANN models, and how the data were processed, organized, and presented. The inclusion of equations to define the process of epitope extraction adds mathematical precision, while the discussion of peptide lengths (9–13 amino acids, consistent with the training data) highlights thoughtful parameterization of the prediction pipeline.

The implementation of the repository is another strength. The description of the web architecture built on Django, SQLite, and Apache demonstrates careful planning and technical competence. The use of UML diagrams and entity-relationship modeling further strengthens the clarity of the development process. The functionalities of the repository are practical and well-designed. Users can query both antigen and epitope tables, filter by organism or protein, and directly download results in FASTA format. The inclusion of “All Organisms” functionality and the ability to navigate between different data layers (proteins, antigens, epitopes) enhances usability. The supplementary material, while brief, provides useful validation by showing the distribution of training epitope lengths, ensuring consistency between model design and application.

From a scientific perspective, the contribution is important. While several general immunoinformatics databases exist (such as IEDB, Vaxign, and PATRIC), none are specifically dedicated to poxviruses. By focusing narrowly on this family of viruses, the authors provide a specialized tool that is immediately relevant for vaccine research in both human and veterinary contexts. The comparison with ViralZone is particularly useful, as it highlights the complementary nature of these resources: ViralZone provides descriptive, curated information about poxvirus biology, while TCEPVDB provides predictive data on antigens and epitopes at proteome scale. This clear distinction underlines the added value of the current work.

It is true that the predictions reported are computational and not experimentally validated. The authors acknowledge this limitation and openly discuss future improvements, including the incorporation of allergenicity and toxicity filters, as well as the expansion of the resource to B-cell epitopes and additional pathogens. These limitations, however, do not diminish the immediate utility of the resource. On the contrary, they highlight the forward-looking nature of the project. The open-access and open data philosophy of the repository means that experimentalists can directly use these predictions to design validation studies, thereby accelerating translational vaccine development. The repository thus serves as a catalyst for further research, bridging computational predictions with experimental immunology.

The manuscript also demonstrates strong collaboration across multiple institutions, with contributions spanning methodology, software implementation, and data analysis. The funding sources, including CIHR and Moderna Global Fellowship support, underline the relevance and credibility of the research. The conflict of interest statement is transparent and does not appear to detract from the scientific integrity of the work.

In conclusion, this manuscript introduces a resource that will be of substantial value to immunologists, virologists, and vaccine developers. It combines methodological rigor, technical innovation, and practical usability in a way that is rare for computational biology tools. The database fills an unmet need, is openly accessible, and is carefully documented. While future expansions and validations will certainly strengthen its impact, the current version is already highly useful and deserves publication.
I therefore recommend acceptance in its present form. The work is a clear contribution to the field of proteomics, immunoinformatics, and reverse vaccinology, and it will be of interest to a broad readership.

Author Response

Comments and Suggestions for Authors

The manuscript entitled “TCEPVDB: Artificial intelligence based proteome wide screening of antigens and linear T-cell epitopes in the poxviruses and the development of a repository” describes the development of a novel database dedicated to poxvirus antigens and epitopes. The authors employed the PoxiPred deep learning framework to systematically screen the proteomes of 37 distinct poxviruses, predicting 3,966 antigens and more than 54,000 linear T-cell epitopes. These predictions were then organized into a comprehensive, open-access repository TCEPVDB that can be freely queried by the scientific community. The work is timely, highly relevant, and represents a significant step forward in applying computational vaccinology to viral pathogens of public health concern.

The article is well structured and clearly written. The introduction provides a strong rationale for the work, situating poxviruses in both historical and modern contexts, from smallpox eradication to the mpox outbreak of 2022. The authors explain the importance of reverse vaccinology and the role of computational methods in accelerating antigen discovery. This sets the stage well for the contribution of TCEPVDB as a specialized resource built specifically for poxviruses. The methods section is transparent and rigorous. The authors describe in detail how proteomes were retrieved from NCBI, how antigens and epitopes were predicted using the PoxiPred DL-ANN models, and how the data were processed, organized, and presented. The inclusion of equations to define the process of epitope extraction adds mathematical precision, while the discussion of peptide lengths (9–13 amino acids, consistent with the training data) highlights thoughtful parameterization of the prediction pipeline.

The implementation of the repository is another strength. The description of the web architecture built on Django, SQLite, and Apache demonstrates careful planning and technical competence. The use of UML diagrams and entity-relationship modeling further strengthens the clarity of the development process. The functionalities of the repository are practical and well-designed. Users can query both antigen and epitope tables, filter by organism or protein, and directly download results in FASTA format. The inclusion of “All Organisms” functionality and the ability to navigate between different data layers (proteins, antigens, epitopes) enhances usability. The supplementary material, while brief, provides useful validation by showing the distribution of training epitope lengths, ensuring consistency between model design and application.

From a scientific perspective, the contribution is important. While several general immunoinformatics databases exist (such as IEDB, Vaxign, and PATRIC), none are specifically dedicated to poxviruses. By focusing narrowly on this family of viruses, the authors provide a specialized tool that is immediately relevant for vaccine research in both human and veterinary contexts. The comparison with ViralZone is particularly useful, as it highlights the complementary nature of these resources: ViralZone provides descriptive, curated information about poxvirus biology, while TCEPVDB provides predictive data on antigens and epitopes at proteome scale. This clear distinction underlines the added value of the current work.

It is true that the predictions reported are computational and not experimentally validated. The authors acknowledge this limitation and openly discuss future improvements, including the incorporation of allergenicity and toxicity filters, as well as the expansion of the resource to B-cell epitopes and additional pathogens. These limitations, however, do not diminish the immediate utility of the resource. On the contrary, they highlight the forward-looking nature of the project. The open-access and open data philosophy of the repository means that experimentalists can directly use these predictions to design validation studies, thereby accelerating translational vaccine development. The repository thus serves as a catalyst for further research, bridging computational predictions with experimental immunology.

The manuscript also demonstrates strong collaboration across multiple institutions, with contributions spanning methodology, software implementation, and data analysis. The funding sources, including CIHR and Moderna Global Fellowship support, underline the relevance and credibility of the research. The conflict of interest statement is transparent and does not appear to detract from the scientific integrity of the work.

In conclusion, this manuscript introduces a resource that will be of substantial value to immunologists, virologists, and vaccine developers. It combines methodological rigor, technical innovation, and practical usability in a way that is rare for computational biology tools. The database fills an unmet need, is openly accessible, and is carefully documented. While future expansions and validations will certainly strengthen its impact, the current version is already highly useful and deserves publication.
 I therefore recommend acceptance in its present form. The work is a clear contribution to the field of proteomics, immunoinformatics, and reverse vaccinology, and it will be of interest to a broad readership.

 Response: We would like to sincerely thank Reviewer 1 for their positive evaluation of our manuscript and for recommending acceptance in its current form. It is encouraging and supporting.  We are grateful for the time and effort they dedicated to reviewing our work.

Reviewer 2 Report

Comments and Suggestions for Authors

The Dutt study aimed to identify antigenic proteins, and more specifically, T-cell epitopes, in poxviruses. To this end, the researchers analyzed 7,185 proteins from 37 different viruses. The research employed a two-step approach: first, they used the PoxiPred predictor to classify which proteins were potential antigens, resulting in 3,966 selected proteins. Next, they focused on predicting T-cell epitopes by extracting subsequences of these antigens based on the typical length (9 to 13 amino acids) of the epitopes used in model training. The central purpose was to map precise immunological targets in poxviruses to aid in the development of vaccines and immunological therapies. The work addresses a specific gap by focusing on predicting T-cell epitopes, which are fundamental to a long-lasting and effective immune response. Many prediction tools are generic, but the authors used a model ("PoxiPred") specifically for poxviruses, considering the variability of their proteins and the fact that different forms of the same protein (proteoforms) can drastically influence immune recognition. Therefore, the originality lies in the application and refinement of immunoinformatics prediction tools to address the complexity of poxviruses in a targeted and large-scale manner, which is not adequately covered by general methods.

Furthermore, it advances by explicitly incorporating the biological complexity of proteoforms and by adopting a statistically rigorous model for predicting T-cell epitopes, considering a range of lengths based on real data. This combination of scale, specialization, and methodological refinement offers a more realistic and detailed immunological mapping for the development of vaccines and therapies. I have suggestions as follows:

(1) This manuscript may be discussed in relation to metrics as shown in references or others in relation to methodology of prediction (ROC AUC, accuracy, precision, etc):

Martinez GS, Dutt M, Kelvin DJ, Kumar A. PoxiPred: An Artificial- Intelligence-Based Method for the Prediction of Potential Antigens and Epitopes to Accelerate Vaccine Development Efforts against Poxviruses. Biology (Basel). 2024 Feb 17;13(2):125. doi: 10.3390/biology13020125. PMID: 38392343; PMCID: PMC10887159.

Souza AS, Amorim VMF, Soares EP, de Souza RF, Guzzo CR. Antagonistic Trends Between Binding Affinity and Drug-Likeness in SARS-CoV-2 Mpro Inhibitors Revealed by Machine Learning. Viruses. 2025 Jun 30;17(7):935. doi: 10.3390/v17070935. PMID: 40733553; PMCID: PMC12298528.

Kacen, A., Javitt, A., Kramer, M.P. et al. Post-translational modifications reshape the antigenic landscape of the MHC I immunopeptidome in tumors. Nat Biotechnol 41, 239–251 (2023). https:// doi.org/10.1038/s41587-022-01464-2

Pritam M. Exploring the whole proteome of monkeypox virus to design B cell epitope-based oral vaccines using immunoinformatics approaches. Int J Biol Macromol. 2023 Dec 1;252:126498. doi: 10.1016/ j.ijbiomac.2023.126498. Epub 2023 Aug 26. PMID: 37640189.

(2) The authors must improve the conclusion, explaining the importance of high-throughput antigen prediction using a specialized tool (PoxiPred), highlighting the main findings and their applications. The authors must consider the biological complexity of the problem.

(3) In addition, I suggest corrections in terminologies and information, for example: "TCEPVDB is composed of two mysql tables..." and "We used the relational structured database system SQLite..." "Despite smallpox being considered an eradicated disease as of August 2025..." The authors must verify this information in WHO website. This disease was eradicated in 1980.”

Author Response

Comments and Suggestions for Authors

The Dutt study aimed to identify antigenic proteins, and more specifically, T-cell epitopes, in poxviruses. To this end, the researchers analyzed 7,185 proteins from 37 different viruses. The research employed a two-step approach: first, they used the PoxiPred predictor to classify which proteins were potential antigens, resulting in 3,966 selected proteins. Next, they focused on predicting T-cell epitopes by extracting subsequences of these antigens based on the typical length (9 to 13 amino acids) of the epitopes used in model training. The central purpose was to map precise immunological targets in poxviruses to aid in the development of vaccines and immunological therapies. The work addresses a specific gap by focusing on predicting T-cell epitopes, which are fundamental to a long-lasting and effective immune response. Many prediction tools are generic, but the authors used a model ("PoxiPred") specifically for poxviruses, considering the variability of their proteins and the fact that different forms of the same protein (proteoforms) can drastically influence immune recognition. Therefore, the originality lies in the application and refinement of immunoinformatics prediction tools to address the complexity of poxviruses in a targeted and large-scale manner, which is not adequately covered by general methods.

Furthermore, it advances by explicitly incorporating the biological complexity of proteoforms and by adopting a statistically rigorous model for predicting T-cell epitopes, considering a range of lengths based on real data. This combination of scale, specialization, and methodological refinement offers a more realistic and detailed immunological mapping for the development of vaccines and therapies. I have suggestions as follows:

(1) This manuscript may be discussed in relation to metrics as shown in references or others in relation to methodology of prediction (ROC AUC, accuracy, precision, etc):

Martinez GS, Dutt M, Kelvin DJ, Kumar A. PoxiPred: An Artificial- Intelligence-Based Method for the Prediction of Potential Antigens and Epitopes to Accelerate Vaccine Development Efforts against Poxviruses. Biology (Basel). 2024 Feb 17;13(2):125. doi: 10.3390/biology13020125. PMID: 38392343; PMCID: PMC10887159.

Souza AS, Amorim VMF, Soares EP, de Souza RF, Guzzo CR. Antagonistic Trends Between Binding Affinity and Drug-Likeness in SARS-CoV-2 Mpro Inhibitors Revealed by Machine Learning. Viruses. 2025 Jun 30;17(7):935. doi: 10.3390/v17070935. PMID: 40733553; PMCID: PMC12298528.

Kacen, A., Javitt, A., Kramer, M.P. et al. Post-translational modifications reshape the antigenic landscape of the MHC I immunopeptidome in tumors. Nat Biotechnol 41, 239–251 (2023). https:// doi.org/10.1038/s41587-022-01464-2

Pritam M. Exploring the whole proteome of monkeypox virus to design B cell epitope-based oral vaccines using immunoinformatics approaches. Int J Biol Macromol. 2023 Dec 1;252:126498. doi: 10.1016/ j.ijbiomac.2023.126498. Epub 2023 Aug 26. PMID: 37640189.

Response: Thank you, per your suggestion, we have discussed these reports in the discussion section (lines 281-295).

“The methodology for the prediction of antigens and epitopes utilized in the present study is solely based on the Poxipred method with ROC AUC metrics, which has been defined as a key indicator of classification performance [20]. Similar trends have already been followed in other related studies. Previously, Souza et al. [33], have significantly applied the precision–recall trade-offs to validate the model reliability in drug-likeness prediction for the screening of SARS-CoV-2 inhibitors. In another recent study, Kacen et al.,[34], highlighted the importance of antigenic landscape modeling with the aid of tumor immunopeptidomics. To predict the effective vaccine candidates for the mpox or other contagious poxviruses, a combined sequence and structure-based approach is required to enhance the accuracy and productivity, as reported by Pritam, [35], in a proteome-wide immunoinformatics study. These reports underscore the value of rigorous evaluation of available epitope prediction tools. In the future, incorporating the different ML-based metrics into the validation of anti-gen and epitopes embedded in the TCEPVDB may give strength to its comparative po-sitioning and significant utilization for the development of effective vaccine candidates against multiple poxviruses. ”

(2) The authors must improve the conclusion, explaining the importance of high-throughput antigen prediction using a specialized tool (PoxiPred), highlighting the main findings and their applications. The authors must consider the biological complexity of the problem.

Response: Thank you for the valuable suggestion. We have significantly improved the conclusion section in the revised version of the manuscript (lines 305-315).

The developed TCEPVDB database is devoted to providing a comprehensive catalogue of a total of 3966 proteins as potential antigen targets and 54291 linear T-cell epitopes from as many as 37 distinct poxviruses. The antigen proteins and linear T-cell epitopes embedded in this database are predicted using the AI-based Poxipred method. The TCEPVDB is a user-friendly database and can be freely accessed using the URL: https://tcepvdb.microbiologyandimmunology.dal.ca/. With further progress in genome sequencing and the AI-based screening of the antigens and epitopes, we anticipate that the number of entries in the TCEPVDB will eventually grow in the upcoming years. Taken together, the information available in the TCEPVDB can be used in efforts of reverse vaccinology, facilitating the rapid development of effective vaccines to tackle poxviruses in a significant manner.  

(3) In addition, I suggest corrections in terminologies and information, for example: "TCEPVDB is composed of two mysql tables..." and "We used the relational structured database system SQLite..." "Despite smallpox being considered an eradicated disease as of August 2025..." The authors must verify this information in WHO website. This disease was eradicated in 1980.”

 Response: Thank you for the suggestion. We have revised the sentences accordingly (lines 44-45, and 171-172).

Reviewer 3 Report

Comments and Suggestions for Authors

The paper is quite interesting in using AI in screening potential antigens and linear T-cell epitopes. However, the results are quite preliminary and should be improved.

Some issues that need to be addressed include:

1. The paper used PoxiPred method for prediction. In this aspect, the paper needs to highlight the difference and novelty of their results compared with Poxipred in terms of accuracy, significance, etc.

2. The paper suffers heavily with lack of experimental validation. Predictions can be easily made, however, knowing their impact can be hard. In this aspect, their predicted results should be compared with available experimental data or literature about epitopes for poxviruses.

3. A lof of predicted epitopes are given, however, the paper needs to establish and streamline which epitopes are biologically important. Information on conservation of epitopes across different poxvirus species can be made. A venn diagram or epitope sequence enrichment can be added to show the relevance of different epitopes across different poxviruses.

4. The paper has a section comparing ViralZone and TCEPVDB. However, this is quite irrelevant. ViralZone hosts information about the virus biology. TCEPVDB is for vaccine design based on antigen and epitope. The authors should have compared their system with similar vaccine design softwares such as Vaxign, to include others. 

5. The paper should have compared their results with two or three more existing softwares for epitope and antigen recognition softwares for poxviruses as a cross-validation or comparison. Analyze differences between the strengths and weaknesses of the new system as compared with already existing software programs.

6. The methods part needs improvement. Especially with the rationale and biological aspect of the development of the system:

 “We ran the models in entire proteins to determine whether they are antigens or not.” 

-What criteria (biological rationale) were used to know whether they are antigens or not? Was it manually done or automated?

“Deep Learning Artificial Neural Network (DL-ANN) models” were used. 

-What is the biological rationale for using DL-ANN model and how did they do the modelling aspect.

“prediction score equal or greater than 0.5 in the sigmoid output layer of their respective DL-ANN models.” were used.

-Was 0.5 used out of convenience? What statistical methods were utilized to establish 0.5 as a benchmark score? Is 50% OK as a threshold, or is it quite conservatively low?

Over-all, the methods section is quite short and does not offer information whether the model and simulations were conducted properly. The author should add relevant information, when necessary.

Minor suggestion:

The paper has a number of typographical errors including August 2026 (line 209), to include others.

Author Response

Comments and Suggestions for Authors

The paper is quite interesting in using AI in screening potential antigens and linear T-cell epitopes. However, the results are quite preliminary and should be improved.

Some issues that need to be addressed include:

1. The paper used PoxiPred method for prediction. In this aspect, the paper needs to highlight the difference and novelty of their results compared with Poxipred in terms of accuracy, significance, etc.

Response: We appreciate the time and effort Reviewer #3 dedicated to reviewing our work. We wanted to clarify that we did not implement any new instance of a predictor. In this work, we used the predictive models enabled by PoxiPred in a large cohort of poxviruses proteomes to predict potential antigens and epitopes. Moreover, the scope of the proposed work is to model in a user-friendly format a database containing the outputs we obtained from executing PoxiPred. In order to make this design choice clear to the readers, we made the goal of the proposed work explicit in the abstract (lines 16-19).

“Here, we propose the modeling of a user-friendly database containing the predicted antigens and epitopes of a large cohort of poxviruses proteome using the existing PoxiPred method for reverse vaccinology of poxviruses.”

 and in the last paragraph of our introduction (lines 80-83).

“In this work, we present the T-Cell Epitopes Poxviruses Database (TCEPVDB). We obtained the protein repertoire of 37 distinct poxviruses and submitted them for vaccine components prediction using the PoxiPred method. The predicted outputs are modelled in a user-friendly database. Here, we document the development stage of TCEPVDB. Users interested in exploring the data of TCEPVDB can freely access the tool at https://tcepvdb.microbiologyandimmunology.dal.ca.”

2. The paper suffers heavily with lack of experimental validation. Predictions can be easily made, however, knowing their impact can be hard. In this aspect, their predicted results should be compared with available experimental data or literature about epitopes for poxviruses.

Response: We thank the reviewer for their feedback. The scope of our work dealt with proposing the modelling of a database containing the outputs of the execution of the PoxiPred model, which was previously published. In the revised MS, we have added the flowchart of the pipeline utilized in this study. We recognize the in-silico identification of components for the reverse vaccinology of poxviruses is a primary data curation step in order to produce candidates that are based on a methodological approach, i.e., the AI-based prediction of epitopes and antigens. We recognize this as a potential limitation of our work and added it in our discussion section (lines 297-301).

“In conclusion, the in-silico step of reverse vaccinology serves primarily as a data curation step, generating preliminary predictions of candidate epitopes and antigens. Their prediction requires subsequent experimental validation and biological interpretation to refine the set of candidates with true potential for incorporation into vaccine designs.”

3. A lot of predicted epitopes are given, however, the paper needs to establish and streamline which epitopes are biologically important. Information on conservation of epitopes across different poxvirus species can be made. A venn diagram or epitope sequence enrichment can be added to show the relevance of different epitopes across different poxviruses.

Response: We appreciate the reviewer’s constructive comment. For each epitope that we modelled in our Database, we checked its presence across each of the poxvirus species we included in this work. This gave us a heatmap (Figure 4) in which we can visualize the epitope conservation and cross-species distribution. We included a new section in our results to highlight this new experiment [lines 133-138 (methodology), 238-250 (results)].

2.5 Conservation of epitopes across poxvirus species

Predicted epitope sequences from each organism were extracted from their files and scanned against the proteome of all other organisms. For each source organism, the number of its epitopes present in each proteome was counted and compiled into a matrix. The resulting matrix was visualized as a heatmap to illustrate epitope conservation and cross-species distribution.

3.3 Conservation of predicted epitopes across poxviruses species

To check the conservation of the predicted epitopes, we investigated the presence or absence of each of the predicted epitopes across the entire protein repertoire of the 37 poxviruses we have data available for at TCEPVDB (Figure 4). The poxviruses ectromelia, horsepox, Mpox, taterapox, vaccinia, variola, camelpox, and cowpox had over one thousand epitopes shared between themselves, attesting for the conservation between these poxviruses which might be exploited in cross-protective vaccine development.

Figure 4. Conservation of epitopes across different poxvirus species.  In Figure 4, we show a heatmap of epitope conservation across different poxvirus species. Each epitope that was predicted by PoxiPred and modelled on TCEPVDB had its presence/absence in the proteome of the poxvirus species considered in this study compiled. The heatmap consists of the absolute presence of epitopes across poxvirus species.

4. The paper has a section comparing ViralZone and TCEPVDB. However, this is quite irrelevant. ViralZone hosts information about the virus biology. TCEPVDB is for vaccine design based on antigen and epitope. The authors should have compared their system with similar vaccine design softwares such as Vaxign, to include others.

Response: We appreciate your valuable suggestion and have removed the comparison between ViralZone and TCEPVDB from the revised manuscript, as recommended. Of note, we would like to clarify that TCEPVDB is a comprehensive database rather than a tool or pipeline, Figure 1 presented a flowchart of the pipeline utilized in the present study.  It integrates proteome-wide predictions of antigens and linear epitopes in poxviruses, generated using our previously published PoxiPred method. To the best of our knowledge, there is currently no other publicly available database that is specifically focused on poxviruses and provides curated access to both antigenic proteins and predicted linear epitopes. We believe TCEPVDB fills a unique gap in the field.

5. The paper should have compared their results with two or three more existing softwares for epitope and antigen recognition softwares for poxviruses as a cross-validation or comparison. Analyze differences between the strengths and weaknesses of the new system as compared with already existing software programs.

Response: Thank you for the valuable suggestion regarding the comparison between existing epitope and antigen prediction tools and TCEPVDB. In response, we have revised the manuscript and added Table 2 for comparing TCEPVDB with representative vaccine design platforms such as Vaxign, VaxiJen, and IEDB's epitope prediction tools. These tools offer broad-spectrum antigen prediction across a plethora of pathogens; they are not specifically tailored for poxviruses. While TCEPVDB distinguishes itself by providing a poxvirus-focused database that integrates proteome-wide predictions of antigenic proteins and linear epitopes using our previously published PoxiPred method (lines 217-222).

6. The methods part needs improvement. Especially with the rationale and biological aspect of the development of the system:

 “We ran the models in entire proteins to determine whether they are antigens or not.”

Response: We appreciate the attention to detail. We wanted to say that we first scanned the entire protein repertoire of the included viruses to determine which proteins are and are not antigens. This has been mended (lines xx).

“First, the antigen prediction model was applied to each individual protein from the dataset”

-What criteria (biological rationale) were used to know whether they are antigens or not? Was it manually done or automated?

Response: We appreciate having the opportunity to clarify. The classification framework of PoxiPred consists of two stages: i) the prediction of antigens, where a protein gets predicted as an antigen or non-antigen. ii) the prediction of epitopes, substrings of a protein (the ones predicted as antigens) get predicted whether they are epitopes or non-epitopes. No biological bias was applied in stages either i or ii. We understand the application of such method to be an agnostic way to curate large blocks of data. We made this design choice cleared in our methodology (lines 95-106).

“PoxiPred[20] was originally developed as an agnostic classification framework for predicting antigens and LTCEs in poxvirus protein datasets, functioning as an early data curation step. For the construction of TCEPVDB, we did not develop new models; instead, we employed the pre-trained Deep Learning Artificial Neural Network (DL-ANN) models for (i) antigen prediction and (ii) LTCE prediction. The models are publicly available at https://github.com/gustavsganzerla/poxipred. Using these existing models, we analyzed the protein repertoire of 37 distinct poxviruses. First, the antigen prediction model was applied to each individual protein from the dataset. Proteins predicted as potential antigens were then fragmented into smaller peptides, which were subsequently evaluated using the LTCE prediction model. In both instances, predictions were considered as positive when the sigmoid output layer of the corresponding pre-trained model produced a score ≥ 0.5.”

“Deep Learning Artificial Neural Network (DL-ANN) models” were used.

Response: We appreciate the comment. We want to emphasize that we used the pre-trained models of PoxiPred to obtain the objects we modelled at TCEPVDB. We have rewritten our methods to make this clear to the readers (lines 95-99).

“PoxiPred[20] was originally developed as an agnostic classification framework for predicting antigens and LTCEs in poxvirus protein datasets, functioning as an early data curation step. For the construction of TCEPVDB, we did not develop new models; instead, we employed the pre-trained Deep Learning Artificial Neural Network (DL-ANN) models for (i) antigen prediction and (ii) LTCE prediction”

-What is the biological rationale for using DL-ANN model and how did they do the modelling aspect.

Response: We appreciate the question. We wanted to clarify that the obtention of predictive models was not part of the scope of this work. Here, we wanted to model a database that consists of objects that were predicted by an existing classification framework. The consolidation of PoxiPred used DL-ANN models. The original publication identified that DL-ANNs would be the most suitable classification method for the problem specified. Six different machine learning models were tested and DL-ANNs outperformed the other five.

“prediction score equal or greater than 0.5 in the sigmoid output layer of their respective DL-ANN models.” were used.

-Was 0.5 used out of convenience? What statistical methods were utilized to establish 0.5 as a benchmark score? Is 50% OK as a threshold, or is it quite conservatively low?

Response: This is an important question. By default, the output of a sigmoid function (the function used in the last layer of the PoxiPred model, ranging from 0 to 1) assigns a positive class (predicted epitope/antigen) for any observation whose prediction score is >=0.5. The closer the score is to 1, the higher the probability of identifying a positive class according to the model. In the web version of our database, we included a column indicating the prediction score to inform users on how certain the models are on the predictions. We have added a passage in our results describing the prediction score rationale (lines 200-204).

“Both the epitope and antigen models in TCEPVDB contain a score attribute, which is defined by the output of the sigmoid function applied at the final layer of the neural networks implemented in PoxiPred. Entries displayed in TCEPVDB correspond to predicted antigens and epitopes with sigmoid scores ≥0.5, reflecting the classification threshold used to determine positive predictions.”

Over-all, the methods section is quite short and does not offer information whether the model and simulations were conducted properly. The author should add relevant information, when necessary.

Response: We appreciate having the opportunity to clarify such matter. The training process of the model is derived from an already existing publication that proposed PoxiPred. The scope of the proposed work is to provide a user-friendly modelling of the outcomes of executing such models on a cohort of proteomes of different poxviruses. We have modified our methodology section to make this message clear to the readers. We are open to any specific suggestions on how to improve the description of the methodology that pertains to the scope of the proposed work.

Minor suggestion:

The paper has a number of typographical errors including August 2026 (line 209), to include others.

Response: We appreciate the detailed analysis. We corrected the typo listed by the reviewer. We have also had a native English speaker with experience in editing to have a final look at our work.

Round 2

Reviewer 3 Report

Comments and Suggestions for Authors

The authors have revised the manuscript accordingly.